# Valgus Arthritic Knee Responds Better to Conservative Treatment than the Varus Arthritic Knee

**DOI:** 10.3390/medicina59040779

**Published:** 2023-04-17

**Authors:** SeungHoon Lee, YunSeong Choi, JaeHyuk Lee, HeeDong Lee, JungRo Yoon, ChongBum Chang

**Affiliations:** 1Department of Orthopaedic Surgery, Veterans Health Service Medical Center, 53 Jinhwangdo-ro 61-gil, Gangdong-gu, Seoul 05368, Republic of Korea; 2Department of Orthopaedic Surgery, Seoul National University Bundang Hospital, Seongnamsi 13620, Republic of Korea; 3Department of Orthopaedic Surgery, Seoul National University College of Medicine, Seoul 03080, Republic of Korea

**Keywords:** varus arthritic knee, valgus arthritic knee, conservative treatment, total knee arthroplasty, varus alignment, valgus alignment, hip-knee-ankle angle, Kaplan–Meier curve, receiver operating characteristic curve, osteoarthritis

## Abstract

*Background and Objectives*: Clinically, it is beneficial to determine the knee osteoarthritis (OA) subtype that responds well to conservative treatments. Therefore, this study aimed to determine the differences between varus and valgus arthritic knees in the response to conservative treatment. We hypothesized that valgus arthritic knees would respond better to conservative treatment than varus arthritic knees. *Materials and Methods*: Medical records of 834 patients who received knee OA treatment were retrospectively reviewed. Patients with Kellgren–Lawrence grades III and IV were divided into two groups according to knee alignment (varus arthritic knee, hip-knee-ankle angle [HKA] > 0° or valgus arthritic knee, HKA < 0°). The Kaplan–Meier curve with total knee arthroplasty (TKA) as an endpoint was used to compare the survival probability between varus and valgus arthritic knees at one, two, three, four, and five years after the first visit. A receiver operating characteristic (ROC) curve was used to compare the HKA thresholds for TKA between varus and valgus arthritic knees. *Results*: Valgus arthritic knees responded better to conservative treatment than varus arthritic knees. With TKA as an endpoint, the survival probabilities for varus and valgus arthritic knees were 24.2% and 61.4%, respectively, at the 5-year follow-up (*p* < 0.001). The thresholds of HKA for varus and valgus arthritic knees for TKA were 4.9° and −8.1°, respectively (varus: area under the ROC curve [AUC] = 0.704, 95% confidence interval [CI] 0.666–0.741, *p* < 0.001, sensitivity = 0.870, specificity = 0.524; valgus: AUC = 0.753, 95% CI 0.693–0.807, *p* < 0.001, sensitivity = 0.753, specificity = 0.786). *Conclusions*: Conservative treatment is more effective for valgus than for varus arthritic knees. This should be considered when explaining the prognosis of conservative treatment for knees with varus and valgus arthritis.

## 1. Introduction

As total knee arthroplasty (TKA) is increasingly recognized as a standard treatment option for end-stage knee osteoarthritis (OA) with widespread acceptance, its burden is expected to increase significantly [1,2,3]. Singh et al. reported that the number of primary TKAs in the United States would increase by 401% by 2040 [1], and Kim et al. estimated that primary TKA costs would increase by 58–114% by 2030 compared to those in 2018 [2]. As this increase in TKA could become a tremendous burden on society, healthcare, and patients, determining which subtype of knee OA responds well to conservative treatment would be clinically beneficial.

Knee OA is comprised of two subtypes: varus and valgus arthritic knees. Wei et al. reported that these subtypes have different etiologic heterogeneities [4], and this suggests that the two types of knee OA may differ in their response to conservative treatment. Valgus arthritic knees account for approximately 10% of the patients undergoing TKA [5], which is less than the percentage of TKA for varus arthritic knees. This is mainly because most patients with knee OA have a varus alignment of the coronal plane, although it may also be because valgus arthritic knees respond better to conservative treatment than varus arthritic knees. However, no clear consensus has been reached on this issue.

The difference between varus and valgus arthritic knees in response to conservative treatment may be related to the difference in the effects of varus and valgus alignment on the knee joint. Previous studies reported that the effect of valgus alignment on the lateral knee joint was less than that of varus alignment on the medial knee joint [6,7,8,9,10,11]. Sharma et al. reported that varus (adjusted odds ratio [OR] 1.49, 95% confidence interval [CI] 1.06, 2.10) but not valgus alignment was associated with incident osteoarthritis [7] and that the rate of cartilage damage was not higher in valgus alignment than in neutral alignment [6]. Johnson et al. reported that among patients with valgus arthritic knees, 71% (20/28) had predominantly medial loading [8]. Thus, knees with mild or moderate valgus deformities are not at a high risk of lateral knee OA or its progression, and conservative treatment may be more effective in the valgus than in the varus at the same degree of alignment. However, whether differences in the effects of varus and valgus alignments on the knee joint lead to differences in the effects of conservative treatments remains unclear.

Therefore, this study aimed to determine the difference between varus and valgus arthritic knees in response to conservative treatment. We hypothesized that valgus arthritic knees would respond better to conservative treatment than varus arthritic knees, and that this would be related to the lesser effect of valgus alignment on the lateral knee joint than that of varus alignment on the medial knee joint.

## 2. Materials and Methods

### 2.1. Participants

The medical records of 24,382 limbs treated for knee OA in a single institution between January 2000 and December 2016 were reviewed retrospectively. The inclusion criteria were as follows: (1) age ≥ 65 years; (2) degenerative knee OA of Kellgren–Lawrence (KL) grades [12] III and IV at the first visit; (3) pain in the knee on a numeric rating scale (NRS, 0–10) of ≥4 [13] at the first visit; and (4) received conservative treatment for at least five years or TKA within five years after the first visit. The exclusion criteria were as follows: (1) a history of previous treatment for knee OA; (2) unavailable medical records and radiographs; (3) bilateral knee OA with overlapping treatment periods; and (4) unsuitability for conservative treatment (abnormal laboratory findings and side effects such as nausea and vomiting). After applying inclusion and exclusion criteria, 834 limbs (349 from male and 485 from female patients) were analyzed (Figure 1). The average age at the first visit was 74 ± 9 years (range, 65–94 years) and the mean body mass index (BMI) was 25.8 ± 3.5 kg/m^2^ (range, 15.1–38.9).

### 2.2. Patient Treatment Protocol

The treatment protocol for the outpatient clinic is shown in Figure 2. All patients who were treated at our knee OA outpatient clinic first received conservative treatment, defined as the avoidance of invasive procedures such as surgery, for at least 3 months by a knee specialist (JRY). Surgical treatment, such as TKA, was considered within 1 month if the NRS did not improve. The NRS score was evaluated by a knee specialist during each outpatient visit. The conservative treatment at our outpatient clinic included education on lifestyle modifications, oral drugs, and intra-articular injections. Education regarding lifestyle modifications included information on weight loss, physical therapy, exercise, and activity modifications. Advice on weight loss was targeted at patients with a BMI ≥ 25. The advice regarding physical therapy/exercise was comprised of teaching exercises that could strengthen the muscles around the knee, such as quad-set exercises, straight leg raises, level walking, and swimming. The advice regarding activity modification was to avoid activities that could worsen arthritis, such as deep squatting, bending, and stair climbing. Oral drugs prescribed to patients included: acetaminophen 300 mg three times daily; non-steroidal anti-inflammatory drugs (Celebrex 200 mg once daily, Acelex 2 mg once daily, naproxen 500 mg twice daily, and pelubiprofen 45 mg twice daily); opioids (tramadol 75 or 150 mg twice daily); and slow-acting drugs for OA (SYSADOA) (JOINS 200 mg thrice daily and shinbaro 600 mg twice daily). Hyaluronic acid (synovian 3 mL once every 6 months), and corticosteroids (triamcinolone 1 mL, at least 2 weeks apart) were considered for intra-articular injection. Blood tests and side effects were evaluated during outpatient follow-ups, and drug treatment was adjusted in cooperation with the specialist in charge when side effects occurred.

### 2.3. Radiographic Examination

Radiographic evaluations were performed at the first outpatient visit, every 6 months, and immediately before TKA. Radiographic evaluations were performed using standing full lower limb anteroposterior (AP) and standing knee AP radiographs, which were obtained using a UT 2000 X-ray machine (Philips Research, Eindhoven, The Netherlands) set to 90 kV and 50 mA. Both standing full lower limb and standing knee AP radiographs were acquired with the patella facing forward. The hip-knee-ankle angle (HKA) was measured on standing full lower limb AP radiographs, and the KL grade was assessed on standing knee AP radiographs. HKA was defined as the angle between the mechanical axes of the tibia and femur. A positive value was assigned to varus knees. The KL grade was used to evaluate the radiographic severity of knee OA. All radiographic images were acquired digitally using a picture archiving and communication system (PACS). Assessments were performed on a 24-inch (61 cm) LCD monitor (T245; Samsung, Seoul, Republic of Korea) in portrait mode using the PACS software. The minimum changes detectable by the software were 0.1° in angle and 0.1 mm in length. The intra- and inter-observer reliability of all radiographic evaluations were evaluated using intraclass correlation coefficients (ICCs). Two orthopedic surgeons (YSC and JHL) independently performed radiographic evaluations. Radiographic evaluations were performed twice at 3-week intervals by each examiner. The ICCs of the intra-and inter-observer reliabilities were >0.9 (range, 0.948–0.998), indicating highly reliable evaluations (Table 1).

### 2.4. Data Analysis

The difference between varus and valgus arthritic knees in response to conservative treatment was assessed by analyzing the survival probability and the effect of varus and valgus alignment on the knee joint. Knee arthritis was defined as a knee with KL grades III and IV; a varus arthritic knee was defined as HKA > 0°, and a valgus arthritic knee was defined as HKA < 0° [14,15,16]. The demographic and radiographic variables of the groups were compared (Table 2). Survival probability was analyzed by comparing the survival probability between the varus and valgus arthritic knees at one, two, three, four, and five years after the first visit. A “surviving knee” was defined as a knee that received only conservative treatment without TKA during the follow-up period. The effects of varus and valgus alignment on the knee joint were analyzed by comparing the threshold of HKA for TKA between varus and valgus arthritic knees using HKA five years after the first visit in patients with only conservative treatment, and HKA immediately preoperatively in patients who underwent TKA within five years of the first visit.

### 2.5. Statistical Analysis

Continuous variables were expressed as mean ± standard deviation or mean with range. Categorical variables were expressed as numbers or percentages. Differences in continuous and categorical variables were analyzed using the Student’s *t*-test and the chi-squared test, respectively. Kaplan–Meier curves were constructed for survival probability, and TKA within five years after the first visit was considered the endpoint. Finally, a receiver operating characteristic (ROC) curve analysis was performed to identify the HKA threshold for TKA. The threshold was calculated using Youden’s J-statistic. The area under the ROC curve (AUC) ranged from 0.5, indicating a test with no accuracy, to 1.0, indicating that the test is perfectly accurate. The minimum accepted threshold for reliability is defined as an AUC of ≥0.7 [17]. An a priori power analysis was not conducted because this would be inappropriate for a retrospective study [18]. Statistical analyses were performed using SPSS (version 26.0; IBM, Armonk, NY, USA). Kaplan–Meier curves were conducted using R software, version 4.1.2 (R Foundation for Statistical Computing, Vienna, Austria; http://www.R-project.org (accessed on 31 March 2023). Statistical significance was set at *p* < 0.05.

## 3. Results

Valgus arthritic knees responded better to conservative treatment than varus arthritic knees. With TKA as an endpoint, the survival probabilities for varus and valgus arthritic knees were 24.2% and 61.4%, respectively, at the five-year follow-up (*p* < 0.001) (Figure 3). In particular, the survival probability of varus arthritic knees within six months of the first visit was markedly lower than that of valgus arthritic knees (Figure 4). The annual survival probabilities of varus and valgus arthritic knees are summarized in Table 3. The patients who underwent TKA for valgus arthritic knees had greater preoperative coronal alignment deformation than those who underwent TKA for varus arthritic knees. The thresholds of HKA for the varus and valgus arthritic knees for TKA were 4.9° and −8.1°, respectively (varus: AUC = 0.704, 95% CI 0.666–0.741, *p* < 0.001, sensitivity = 0.870, specificity = 0.524; valgus: AUC = 0.753, 95% CI 0.693–0.807, *p* < 0.001, sensitivity = 0.753, specificity = 0.786) (Figure 5).

## 4. Discussion

The most important finding of this study was that conservative treatment was more effective in valgus arthritic knees than in varus arthritic knees. This can be explained by the fact that valgus arthritic knees have a higher five-year survival probability than varus arthritic knees, and the effect of alignment on the knee joint was lower in valgus arthritic knees than in varus arthritic knees.

To the best of our knowledge, no consensus has been reached regarding the differences between varus and valgus arthritic knees in responsiveness to conservative treatment. This study demonstrated that the survival probability of valgus arthritic knees was higher than that of varus arthritic knees using Kaplan–Meier curves. However, this study did not include high tibial osteotomy, unicompartmental knee arthroplasty, or cartilage regeneration surgery as endpoints. As these surgeries were mainly performed on varus arthritic knees, the survival probability of valgus arthritic knees would have been much higher if these surgeries had also been included in this study. TKA is generally successful in reducing pain and improving function in patients with knee OA [19]. However, conservative treatment has become increasingly important because the socioeconomic burden of TKA is expected to increase [1,2,3], and approximately 20% of patients remain dissatisfied after undergoing TKA [20]. Although many successful conservative treatments have been reported [21], the knee OA subtype that responds well to these conservative treatments remained unknown. Clinicians treating patients with knee OA struggled to administer effective conservative treatments to delay TKA for as long as possible. Hence, our findings are meaningful in that we demonstrated that valgus arthritic knees responded better to conservative treatment than varus arthritic knees.

Another notable finding of this study was that the coronal alignment deformation that required TKA was more severe in valgus than in varus arthritic knees. Previous studies suggested that the effect of valgus alignment on the lateral knee joint is less than that of varus alignment on the medial knee joint. Andriacchi and Kutzner et al. reported that the medial compartment absorbs 60–70% of the compressive force under an HKA angle of 0°, and a slight valgus alignment of 3° to 4° leads to an equal distribution of force [22,23]. In addition, Brouwer and Sharma et al. reported that varus alignment is associated with the development and progression of medial knee OA, whereas valgus alignment is related only to the progression of lateral knee OA [7,9,11]. Sharma et al. reported no significant difference in the incidence of lateral cartilage defects during the 30-month follow-up between valgus and neutral alignment [6]. Felson et al. reported that the ground reaction force vector, which extends from the foot’s center of pressure to the body’s center of mass, passes medial to the knee in many patients with valgus alignment [24]. However, we cannot conclude from the findings of previous studies that the difference between varus and valgus arthritic knees in response to conservative treatment is related to the difference in the effects of varus and valgus alignment on the knee joint. Therefore, our findings have clinical significance in that they showed that differences in the effects of varus and valgus alignments on the knee joint can lead to differences in response to conservative treatment between varus and valgus arthritic knees. In other words, the effect of valgus alignment on the lateral compartment seems smaller than that of varus alignment on the medial compartment, which might be one reason why the valgus arthritic knee responded more favorably to conservative treatment than the varus arthritic knee. This study is unique in that the effect of alignment on the knee joints was evaluated using the threshold of HKA for TKA rather than radiography, magnetic resonance imaging, and gait analyses used in previous studies. This method not only evaluated the effect of alignments on the knee joint but also confirmed whether differences in these effects eventually affect treatment. Our findings may serve as a basis for elucidating the prognosis of conservative treatment in patients with varus and valgus arthritic knees.

Anatomical and kinematic differences may be another reason for the difference in response to conservative treatment between varus and valgus arthritic knees. The lateral meniscus is smaller, thinner, and more mobile than the medial meniscus [25,26]. This anatomical feature of the lateral meniscus makes it less susceptible to damage than the medial meniscus. Healthy knees show a specific degree of femoral lateral rollback and medial pivot [27]. That is, the medial femoral condyle shows less anteroposterior translation than the lateral femoral condyle, and these kinematics may cause greater contact stress on the medial tibial plateau than on the lateral tibial plateau, which may make the cartilage of the lateral compartment less vulnerable to damage than that of the medial compartment. However, this has not yet been clarified; therefore, biomechanical studies are needed.

This study has some limitations. First, the results of this study cannot be generalized because of the diversity of conservative treatments. It is unclear whether the same results will be obtained with conservative treatments other than those used in our study. However, the oral drugs and injection therapies in this study have been used in many hospitals. Second, the study had a relatively short follow-up period of five years. A prospective study with long-term follow-up is therefore required to obtain more comprehensive data. Third, the single institution in this study was a hospital where the medical expenses charged to patients were fairly low; therefore, patients could easily undergo TKA. Most patients who underwent TKA switched from conservative treatment to TKA within the first six months for both varus and valgus arthritic knees. This may be related to the institutions’ fairly low medical expenses. However, as this was a common concern in both groups, it did not significantly affect the result. Fourth, this study also included patients with obesity, with a BMI > 30 (wanted 13.7%, unwanted 10.1%, total 12.7%). Since obesity can cause the progression of knee arthritis [28], the proportion of patients receiving conservative treatment and TKA can vary depending on the proportion of enrolled patients with obesity. However, since obesity has a high prevalence, accounting for 23.2% of the causes of TKA in the United States [29], it is more appropriate to include obesity in the analysis. In addition, no statistically significant difference in BMI was found between groups, and obesity was a common condition in both groups. Therefore, the result that conservative treatment is more effective in the valgus arthritic knee was not significantly affected by obesity.

## 5. Conclusions

The conservative treatment was more effective for valgus arthritic knees than for varus arthritic knees. However, the present cohort of patients needs to be followed-up to determine whether they will continue to demonstrate the same results over time. Our findings should be considered when explaining the prognosis of conservative treatment in patients with varus and valgus arthritic knees.

## Figures and Tables

**Figure 1 medicina-59-00779-f001:**
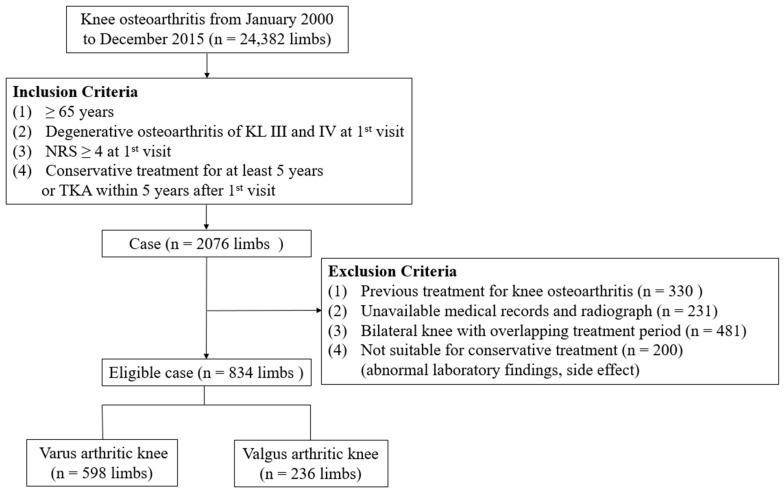
Flow diagram illustrating patient enrollment. n, number; KL, Kellgren–Lawrence; NRS, numeric rating scale; TKA, total knee arthroplasty.

**Figure 2 medicina-59-00779-f002:**
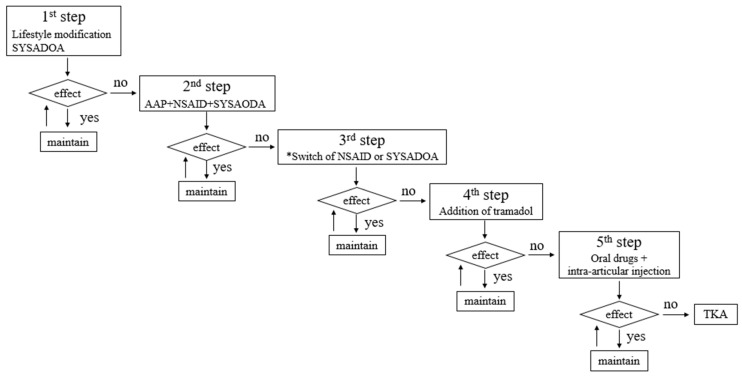
Diagrammatic representation of the treatment protocol. The effect was evaluated using a NRS. If the NRS remained >4 after 2–3 weeks, the next step was performed. (NRS: Higher scores indicate more pain.) * Switching a specific drug was based on previous exposure to specific NSAIDs and SYSADOAs. NRS, numeric rating scale; NSAID, non-steroidal anti-inflammatory drug; SYSADOA, slow-acting drug for osteoarthritis; AAP, acetaminophen; TKA, total knee arthroplasty.

**Figure 3 medicina-59-00779-f003:**
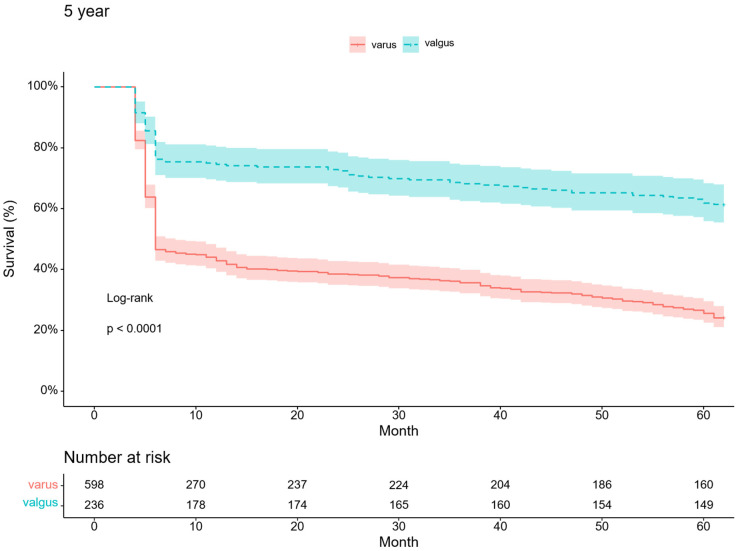
Five-year Kaplan–Meier survivorship curve for varus and valgus arthritic knees.

**Figure 4 medicina-59-00779-f004:**
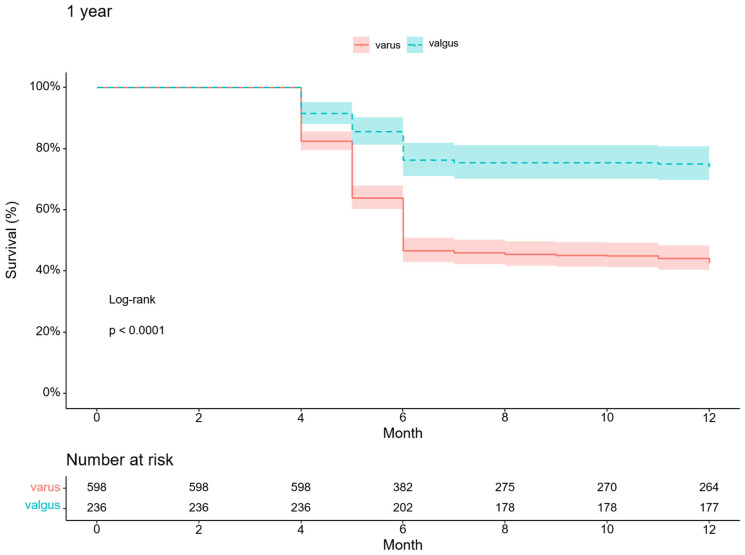
One-year Kaplan–Meier survivorship curve for varus and valgus arthritic knees.

**Figure 5 medicina-59-00779-f005:**
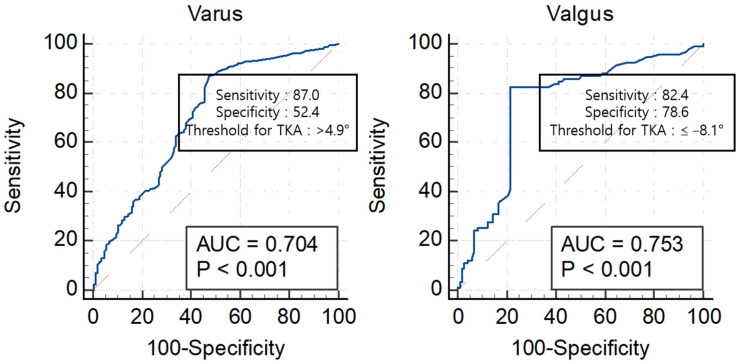
ROC curve for TKA of varus and valgus arthritic knees using the HKA. ROC, receiver operating characteristic; TKA, total knee arthroplasty; AUC, area under the ROC curve; HKA, hip–knee–ankle angle.

**Table 1 medicina-59-00779-t001:** Intra-observer and inter-observer reliability of radiographic evaluations.

	Intra-Observer Reliability	Inter-Observer Reliability	*p* Value
HKA	0.952 (0.910–0.975)	0.948 (0.900–0.973)	*p* < 0.001
KL grade	0.998 (0.997–0.999)	0.973 (0.948–0.986)	*p* < 0.001

HKA, hip–knee–ankle angle; KL, Kellgren Lawrence.

**Table 2 medicina-59-00779-t002:** Patient demographics and radiographic variables (varus arthritic knee vs. valgus arthritic knee).

1st Visit	Varus Arthritic Knee (n = 598)	Valgus Arthritic Knee (n = 236)	*p* Value
**Demographic variable**			
Age (years)	74.5 ± 8.4	73.3 ± 9.4	0.400 *
Gender (female/male)	343/255	142/94	0.154 †
Height (cm)	159.5 ± 9.0	160.2 ± 8.8	0.699 *
Weight (kg)	65.5 ± 12.0	67.0 ± 12.0	0.113 *
BMI (kg/m^2^)	25.7 ± 3.5	26.1 ± 3.5	0.100 *
Laterality (Rt/Lt)	288/310	124/111	0.138 †
Onset of symptom (months)	34.7 ± 14.3	36.1 ± 25.3	0.897 *
NRS (scale)	6.7 ± 2.3	6.1 ± 2.1	0.689 *
**Radiographic variable**			
KL grade (III/IV)	380/218	155/81	0.563 *
HKA	6.4 ± 4.8	−6.5 ± 3.7	<0.001 *

Continuous variables are expressed as means ± standard deviation. Categorical variables are presented as number. BMI, body mass index; Rt, right; Lt, left; NRS, numeric rating scale; KL, Kellgren Lawrence; HKA, hip–knee–ankle angle. * Derived using Student’s *t*-test. † Derived using chi-square test. NRS: Higher scores indicate more pain.

**Table 3 medicina-59-00779-t003:** Survival probability of varus and valgus arthritic knee.

Survival Probability					
Arthritic Knee	1-year	2-year	3-year	4-year	5-year
Varus (n = 598) (n, %)	257 (43)	231 (38.6)	214 (35.8)	189 (31.6)	145 (24.2)
Valgus (n = 236) (n, %)	176 (74.6)	171 (72.5)	161 (68.2)	154 (65.3)	145 (61.4)
*p* value	<0.001	<0.001	<0.001	<0.001	<0.001

n, number.

## Data Availability

Not applicable.

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
