# Peer review of "Valgus Arthritic Knee Responds Better to Conservative Treatment than the Varus Arthritic Knee"

_medicina, 2023, doi:10.3390/medicina59040779_

Round 1

Reviewer 1 Report

General comments:

In this study, arthritic knees of both the varus and valgus types were given conservative treatment, and the researchers compared the results. When evaluating the efficacy of conservative treatment for arthritic knees in valgus and varus positions, the survival probability and the HKA threshold were used as measures of effectiveness. In general, this is not an easy task because it requires the collection of five years' worth of longitudinal data with a sizable sample size. However, the written quality of this paper has room for improvement, which would allow for a significant increase in the impact of this paper. It is not clear why the authors believe that conservative treatment has different effects in varus and valgus arthritic knees, why the valgus arthritic knees would expect to have better outcomes than the varus arthritic knees, or how this knowledge can translate or improve upon current clinical treatment. I would recommend that the authors revise the manuscript to: 1. improve the literature review by providing some potential reasons for why the conservative treatment can cause different effect on varus and valgus arthritic knees; and 2. have a paragraph in the introduction that explains why knowing the differences is important in the current clinical treatment. 3.

Specific comments: 

- Please explain what this paper means by the term "conventional treatment" and provide an explanation as to why this treatment might have a different effect on varus and valgus arthritic knees. This is the primary reason why the authors decided to write this article, and it is necessary to provide specifics regarding the literature review in order to emphasize the research question. The authors, on the other hand, only spend two sentences discussing the problem in the introduction. Previous studies have demonstrated that varus alignment is associated not only with the progression of medial knee OA but also with its development [3, 4, 17, 18]. On the other side, previous studies have demonstrated that the effect of valgus alignment on the lateral knee joint is less than that of varus alignment on the medial knee joint [16]. I would appreciate it if you could go into more detail about these papers.

- It is not completely clear where the hypothesis got its start. It was hypothesized by the authors that varus arthritic knees would respond less favorably to conservative treatment than valgus arthritic knees would. I would guess that it could be based on the findings of paper 16. If this is the case, could you please call attention to it and give me more information about this paper?

- There is also a lack of clarity regarding the significance of understanding the distinction between varus and vaguls in the context of conservation treatment. In the following paragraph, please address the aforementioned question and explain how the findings relate to clinical treatment.

Reviewer 2 Report

Important data is provided that is applicable to the current practice of the arthritic knee.

The following points should be addressed before the paper can be considered for publication:

Background and Objectives:

what is the hypothesis? What is the current practice?

Introduction:

“Meanwhile” – consider omitting.

Subjects:

“(BMI) was 25.8 ± 3.5 kg/m2 (range, 15.1–38.9)” – what is the percentage with BMI>30 (obesity) , overall and in each group?  Discuss this point in relation to the results.

Treatment protocol:

“TKA, was considered within 1 month if the NRS did not improve”  -why? Empirically or for other reason

“Hyaluronic acid (synovian 3 ml once every 6 months)” – how these fits to “TKA, was considered within 1 month if the NRS did not improve” 

Figure 3:

Meaning of varus / valgus censored?

Results:

In the survivorship analysis, 95% confidence intervals for each point should be given for a more meaningful interpretation. Providing a “life table” for the survivorship analysis is desirable.

Figure 4

For better understanding by the reader, change the “Criterion” to Threshold for TKA

Round 2

Reviewer 1 Report

The authors have addressed all my comments. I don't have further questions.   

Reviewer 2 Report

Suggestions were addressed - can be published.